# A luminous blue kilonova and an off-axis jet from a compact binary merger at $z = 0.1341$

E. Troja [1,2], G. Ryan [3], L. Piro[4], H. van Eerten[5], S.B. Cenko [2,3], Y. Yoon[6], S.-K. Lee[6], M. Im[6], T. Sakamoto[7], P. Gatkine[1], A. Kutyrev[1,2] & S. Veilleux[1,3]

The recent discovery of a gamma-ray burst (GRB) coincident with the gravitational-wave (GW) event GW170817 revealed the existence of a population of low-luminosity short duration gamma-ray transients produced by neutron star mergers in the nearby Universe. These events could be routinely detected by existing gamma-ray monitors, yet previous observations failed to identify them without the aid of GW triggers. Here we show that GRB150101B is an analogue of GRB170817A located at a cosmological distance. GRB150101B is a faint short burst characterized by a bright optical counterpart and a long-lived X-ray afterglow. These properties are unusual for standard short GRBs and are instead consistent with an explosion viewed off-axis: the optical light is produced by a luminous kilonova, while the observed X-rays trace the GRB afterglow viewed at an angle of ~13°. Our findings suggest that these properties could be common among future electromagnetic counterparts of GW sources.

[1] Department of Astronomy, University of Maryland, College Park, MD 20742-4111, USA. [2] Astrophysics Science Division, NASA Goddard Space Flight Center, 8800 Greenbelt Rd, Greenbelt, MD 20771, USA. [3] Joint Space-Science Institute, University of Maryland, College Park, MD 20742, USA. [4] INAF, Istituto di Astrofisica e Planetologia Spaziali, via Fosso del Cavaliere 100, 00133 Rome, Italy. [5] Department of Physics, University of Bath, Claverton Down, Bath BA2 7AY, UK. [6] Center for the Exploration for the Origin of the Universe, Astronomy Program, Department of Physics and Astronomy, Seoul National University, 1 Gwanak-ro, Gwanak-gu, Seoul 08826, South Korea. [7] Department of Physics and Mathematics, Aoyama Gakuin University, 5-10-1 Fuchinobe, Chuo-ku, Sagamihara-shi Kanagawa 252-5258, Japan. Correspondence and requests for materials should be addressed to E.T. (email: eleonora.troja@nasa.gov)

The second run (O2) of Advanced LIGO and Advanced Virgo led to the breakthrough discovery of the first GW signal[1] from a neutron star (NS) merger coincident with the short duration GRB170817A[2–4], at a distance of 40 Mpc. In several aspects GRB170817A differs from the garden-variety short GRBs observed at cosmological distances. It is a low-luminosity burst with a relatively soft spectrum[4], followed by a bright and short-lived quasi-thermal emission (kilonova[5,6]) peaking at optical and infrared wavelengths[7–10], and a delayed long-lived non-thermal emission (afterglow) visible from X-rays to radio energies[8,11,12]. Throughout the paper we generically refer to the event as GW170817, and specifically refer to AT2017gfo when discussing the kilonova emission, whereas we use GRB170817A when discussing the GRB and afterglow properties. All these nomenclatures refer to different aspects of the same astrophysical event.

The characteristics of GW170817 might explain why similar events eluded identification thus far. Short GRB localizations are mainly based on rapid observations, typically taken by NASA's Neil Gehrels *Swift* Observatory within ~100 s after the burst. At these early times, the UV/optical emission from the kilonova might not be detectable yet, and the delayed onset of the X-ray emission prevents a rapid localization with the *Swift* X-Ray Telescope (XRT). Within the *Swift* sample of ~100 short GRBs, ~30 bursts have no counterpart at longer wavelengths. These events are not localized with enough precision to confidently determine their host galaxy and distance

scale. It is therefore plausible that a few bursts like GRB170817A had already been detected in the nearby Universe but remained unidentified due to the lack of a precise localization. However, within the sample of short GRBs localized by *Swift*, we identified an event which resembles the properties of GRB170817A. The short GRB150101B was fortuitously localized thanks to its proximity to a low-luminosity AGN, initially considered as the candidate X-ray counterpart. The detection of this X-ray source triggered a set of deeper observations at X-ray and optical wavelengths, which uncovered the true GRB afterglow. The properties of GRB150101B are non-standard and, in particular, its X-ray and optical counterparts are brighter and longer-lived than those of other cosmological short GRBs. We show that these observations could be explained by a scenario similar to the one proposed for GW170817 in which the optical emission is powered by a bright blue kilonova[8,9,13], and the X-rays trace the GRB afterglow viewed off-axis[8,12,14].

## Results

**Comparison with other short GRBs.** GRB150101B stands out of the short GRB sample for its prompt emission and afterglow properties. Its gamma-ray phase is weak and very short in duration (~20 ms, Fig. 1), an order of magnitude shorter than GRB170817A. The burst fluence is $\sim 10^{-7}$ erg cm$^{-2}$ (10–1000 keV) corresponding to a total isotropic-equivalent gamma-ray

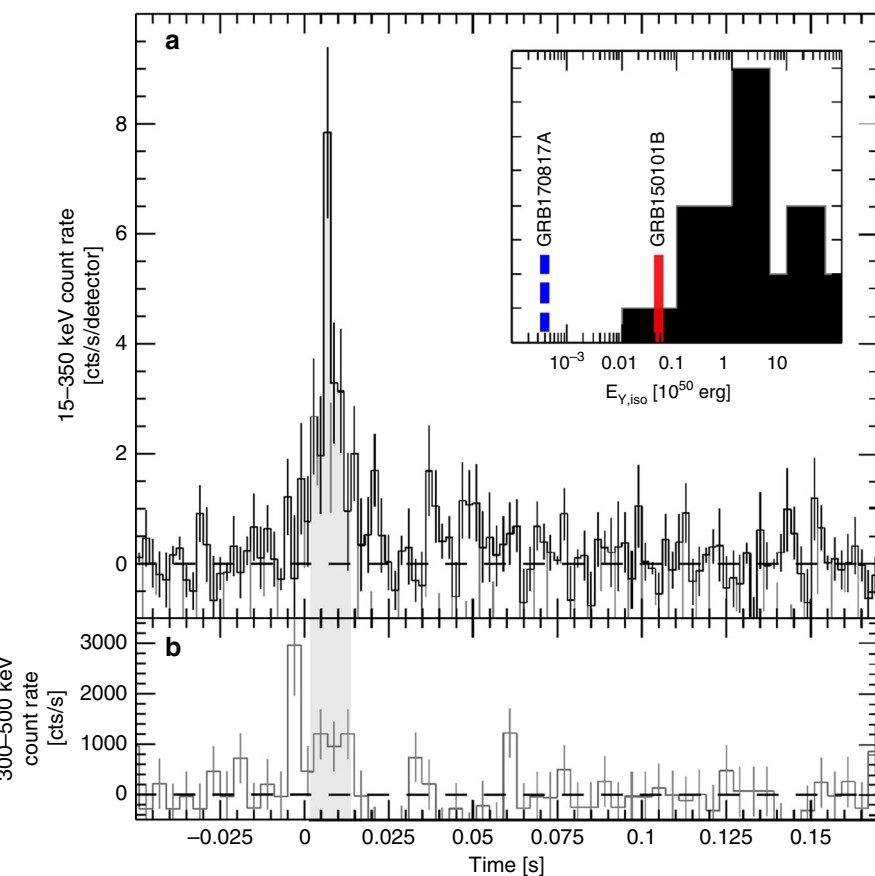

**Fig. 1** Prompt phase of GRB15010B. **a** Gamma-ray light curve of GRB150101B as seen by *Swift* BAT. The time bin is 2 ms. The background contribution is subtracted through the mask-weighting procedure. The shaded area shows the duration $T_{90} \sim 12$ ms of the emission in BAT. Vertical error bars are 1σ. The inset shows the distribution of isotropic-equivalent gamma-ray energy for short GRBs detected by *Swift*[15]. The positions of GRB150101B and, for comparison, GRB170817A are marked by the vertical lines. **b** Background subtracted gamma-ray light curve of GRB150101B above 300 keV, as seen by *Fermi* GBM. The time bin is 4 ms. It shows that at high energies the prompt phase starts a few milliseconds earlier

energy release $E_{\gamma,\text{iso}}$ ~$4.7 \times 10^{48}$ erg at $z = 0.1341$ (see Section Environment), one of the lowest ever detected by *Swift*[15] (Fig. 1, inset). Since this event was not found by the on-board software, no prompt localization was available. Follow-up observations started with a delay of 1.5 days, after a ground-based analysis found the transient gamma-ray source in the *Swift* data. Nonetheless, bright optical and X-ray counterparts were found. This is highly unusual as delayed follow-up observations of short GRBs typically fail to detect the counterpart, especially in the case of weak gamma-ray events.

In the standard GRB model, the broadband afterglow emission is produced by the interaction of the relativistic fireball with the ambient medium[16]. It is therefore expected that the afterglow brightness depends, among other variables, on the total energy release of the explosion[17]. Indeed, typical GRB afterglows are found to be correlated with the total energy radiated in the gamma-rays, being brighter afterglows associated on average to the most luminous GRBs. This is valid for both long duration and short duration bursts[18,19]. In this context, two surprising features of GRB170817A were that, despite its weak gamma-ray emission, the burst was followed by a bright optical transient[20,7], AT2017gfo, and a long-lived X-ray afterglow[8,12]. The observed optical luminosity of AT2017gfo ($L_{\text{pk}}$ ~ $10^{41}$ erg s$^{-1}$) lies within the range of optical afterglows from short GRBs ($10^{40}$ erg s$^{-1}$ < $L_{\text{opt}}$ < $10^{44}$ erg s$^{-1}$ at 12 h). However, when normalizing for the gamma-ray energy release, the optical emission stands out of the afterglow population (Fig. 2a). Indeed, this early UV/optical component is widely interpreted as the kilonova emission from the merger ejecta[8,9,13], and its luminosity is not directly related to the gamma-ray burst as in the case of a standard afterglow. The optical afterglow of GRB170817A was instead much fainter than AT2017gfo at early times and became visible >100 days after the merger[21]. Figure 2a singles out GRB150101B as another event with an optical counterpart brighter than the average afterglow population. Its luminosity ($L_{\text{opt}}$ ~ $2 \times 10^{41}$ erg s$^{-1}$ at 1.5 d) is ~2 times brighter than AT2017gfo at the same epoch and appears to decay at a slower

rate (0.5 ± 0.3 mag d$^{-1}$), although we caution that residual light from the underlying galaxy may be affecting this estimate.

Figure 2b shows the X-ray light curves of short GRBs normalized by the isotropic-equivalent gamma-ray energy release. In this case too, the X-ray counterparts of GRB170817A and GRB150101B differ from the general population of short GRBs and are consistent with the predictions of off-axis afterglows[22]. In the off-axis scenario, the post-peak afterglow reveals the total blastwave energy and is therefore as bright as standard on-axis afterglows at a similar epoch, whereas only a small fraction of the total prompt energy is visible in gamma-rays. Off-axis afterglows are therefore expected to have a $L_X/E_{\gamma,\text{iso}}$ ratio higher than the average population of bursts seen on-axis, as observed for GRB150101B and GRB170817A.

**Temporal analysis.** The earliest follow-up observations of GRB150101B were performed by the *Swift* satellite starting 1.5 days after the burst. *Swift* monitoring lasted for 4 weeks and shows a persistent X-ray source (Fig. 3, top panel). The study of GRB150101B at X-ray energies is complicated by its proximity to a low-luminosity AGN, which contaminates the *Swift* measurements. Observations with the *Chandra X-ray Observatory* (PI: E. Troja, A. Levan) were critical to resolve the presence of the two nearby sources, and to characterize their properties. The brighter X-ray source, coincident with the galaxy nucleus, shows no significant flux or spectral variations between the two epochs (7.9 d and 39.6 d after the burst). Its observed flux is ~$3.7 \times 10^{-13}$ erg cm$^{-2}$ s$^{-1}$ in the 0.3–10 keV energy band, thus accounting for most of the emission measured by *Swift*. The fainter X-ray source, coincident with the position of the GRB optical counterpart[23], instead dropped by a factor of 7 between the two epochs (Fig. 3, bottom panel). By assuming a power-law decay, $F_X \propto t^{-\alpha}$, we derive a temporal decay slope $\alpha \sim 1.2$, typical of standard afterglows. A previous study of this event[23], based only on the *Chandra* dataset, used a simple power-law decay to describe the afterglow temporal evolution from early (~1.5 d) to late (~40 d) times. However, Fig. 3 (top panel) shows that such decay (dashed

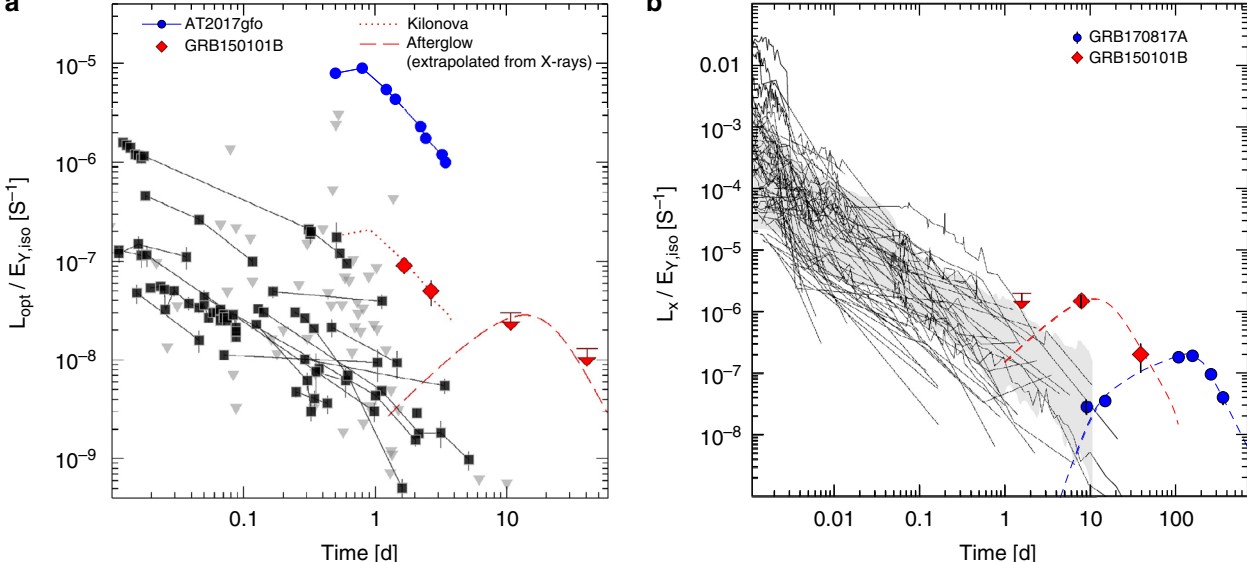

**Fig. 2** Comparison of GRB150101B with GW and GRB afterglows. **a** Optical light curves of short GRBs normalized to the observed gamma-ray energy release. Downward triangles are 3 σ upper limits. Vertical error bars are 1 σ. The optical afterglow of GRB170817A became visible >100 days after the merger[21] and is not reported in the plot. **b** X-ray light curves of short GRBs normalized to the observed gamma-ray energy release. The shaded area shows the 68% dispersion region. In both cases, the counterparts of GW170817 (AT2017gfo and GRB170817A) and GRB150101B stand out of the sample of standard afterglows. They can be described as a bright kilonova (dotted lines) followed by a late-peaking off-axis afterglow (dashed lines)

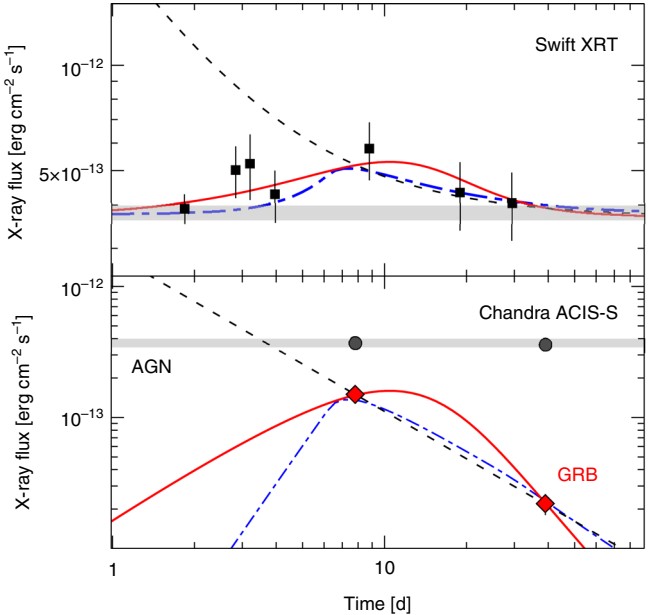

**Fig. 3** X-ray observations of GRB150101B. *Swift* observations (top panel) found a single steady X-ray source. *Chandra* observations (bottom panel) revealed the presence of two nearby X-ray sources: a brighter constant source at the location of the galaxy nucleus (AGN), and a fainter variable source at the location of the GRB optical counterpart. Its X-ray behavior is consistent with a standard power-law decay (dashed line) or a rising afterglow from either an off-axis jet (solid line) or a cocoon (dot-dashed line). However, the superposition of a constant source with a power-law model violates the early *Swift* observations. These are instead well described by the superposition of a constant source with a late-peaking afterglow. Vertical error bars are 1σ. The horizontal gray line shows the average flux level of the AGN

line) would violate the early *Swift* measurements, not considered in past analyses.

The flat *Swift* light curve, although dominated by the AGN contribution, provides an important indication on the behavior of the early GRB afterglow, which had to remain sub-dominant over the observed period. Figure 3 shows that this is consistent with the onset of a delayed afterglow, as observed for GRB170817A[8]. Two leading models are commonly adopted to describe the broadband afterglow evolution of GRB170817A: a highly relativistic structured jet seen off-axis[8,12,14], and a choked jet with a nearly isotropic mildly relativistic cocoon[11,24]. We fit both models to the GRB150101B afterglow with a Bayesian MCMC parameter estimation scheme, using the same priors and afterglow parameters as in ref. [12]. For the structured jet, we assumed that the energy follows a Gaussian angular profile $E(\theta) = E_0 \exp(-\theta)-/2\theta_c{}^2)$ where $\theta_c$ is the width of the energy distribution. The fit results are summarized in Table 1 and shown in Fig. 3 as a solid line (off-axis jet), and a dot-dashed line (cocoon). These models can reproduce both the *Chandra* data (bottom panel) and the *Swift* light curve (top panel). In the cocoon model, the predicted post-peak temporal slope is $\alpha \sim 1.0$, consistent with the result from the simple power-law fit. This implies that the afterglow peak must precede the first *Chandra* observation at 8 d, although not by much as the *Swift* light curve constraints $t_{pk} \gg 1$ d. In the jet scenario, the post-peak temporal decay tends to $\alpha \sim 2.5$, and constrains the range of possible peak times to $t_{pk} \sim 10{-}15$ d.

Other scenarios, such as a long-lasting (~8 d) X-ray plateau followed by a standard afterglow decay, are consistent with the

*Swift* and *Chandra* constraints. However, the required timescales far exceed the typical lifetime ($<10^4$ s) of X-ray plateaus and would not follow the time-luminosity relation[25] usually observed in GRB afterglows.

**Spectral analysis.** The afterglow X-ray spectrum is well described by a simple power-law, $F_\nu \propto \nu^{-\beta}$, with spectral index $\beta = 0.64 \pm 0.17$ (68% confidence level). This implies a non-thermal electron energy distribution with power law slope $p = 2.28 \pm 0.34$ seen below the cooling break, similar to GRB170817A[12,26]. The first *Swift* observation at 1.5 days is likely dominated by the AGN contribution (Fig. 3) and can place a 3σ upper limit on the early X-ray afterglow of ~$1.5 \times 10^{-13}$ erg cm$^{-2}$ s$^{-1}$ (Methods section), if one assumes the same spectral shape as the later *Chandra* epoch. This is a plausible assumption as, for example, continued monitoring of GRB170817A shows no significant variations in its afterglow spectral shape during the first 12 months[12,26].

In Fig. 4, we show the spectral energy distribution of the GRB counterpart. At early times, the optical luminosity exceeds the afterglow extrapolation based on the X-ray limits. This optical excess is consistent with the emergence of a kilonova slightly brighter (by a factor of ~2) than AT2017gfo. Given the limited dataset, we cannot exclude that the optical excess is due to an intrinsic variability of the afterglow (e.g., flares). However, these types of chromatic features are usually observed at X-ray rather than at optical wavelengths, typically occur within a few hours after the GRB, and are more frequent in long GRBs than in short GRBs[27,28]. The luminosity and timescales of the observed optical excess more naturally fit within the kilonova scenario[6,29,30]. The observed color ($r{-}J < 1.2$ at 2.5 d) is somewhat bluer than the color of AT2017gfo at the same epoch, possibly indicating a higher temperature of the ejecta. A slower cooling rate is also consistent with the shallower temporal decay of the optical light.

**Table 1 Afterglow parameters for GRB150101B**

| Afterglow parameter[a] | Value |
|---|---|
| *Structured jet* | |
| $p$ | 2.3 |
| $n$ ($10^{-2}$ cm$^{-3}$) | 7 [0.1, 60] |
| $E_{k,iso}$ ($10^{53}$ erg) | 1.6 [0.6, 9] |
| $\varepsilon_e$ | >0.1 |
| $\varepsilon_B$ ($10^{-5}$) | 3 [1, 30] |
| $\theta_c$ | 3 [1, 5] |
| $\theta_v$ | 13 [8, 20] |
| $\theta_w$ | 28 [14, 53] |
| *Isotropic cocoon* | |
| $p$ | 2.3 |
| $n$ ($10^{-2}$ cm$^{-3}$) | 0.003 [$10^{-6}$, 2] |
| $E_{k,iso}$ ($10^{53}$ erg) | >0.1 |
| $\varepsilon_e$ | >0.01 |
| $\varepsilon_B$ ($10^{-5}$) | 89 [4, 6000] |
| $M_{ej}$ ($M_{sun}$) | <0.004 |
| $u_{min}$ | 8 [0.8, 28] |
| $u_{max}$ | 20 [6, 72] |
| $k$ | 4 [1, 7] |

Numbers in parenthesis indicate the 68% uncertainty interval
[a]Our models are characterized by standard afterglow parameters: spectral index $p$ (held fixed), circumburst density $n$, isotropic-equivalent blastwave energy $E_{k,iso}$, and shock microphysical parameters $\varepsilon_e$ and $\varepsilon_b$. The structured jet model is further defined by the three angular parameters: the Gaussian width of the energy distribution $\theta_c$, the viewing angle $\theta_v$, and the wings truncation angle $\theta_w$. The cocoon model is defined by the mass of relativistic ejecta $M_{ej}$, their maximum $u_{max}$ and minimum $u_{min}$ velocity, and their energy profile with power law slope $k$

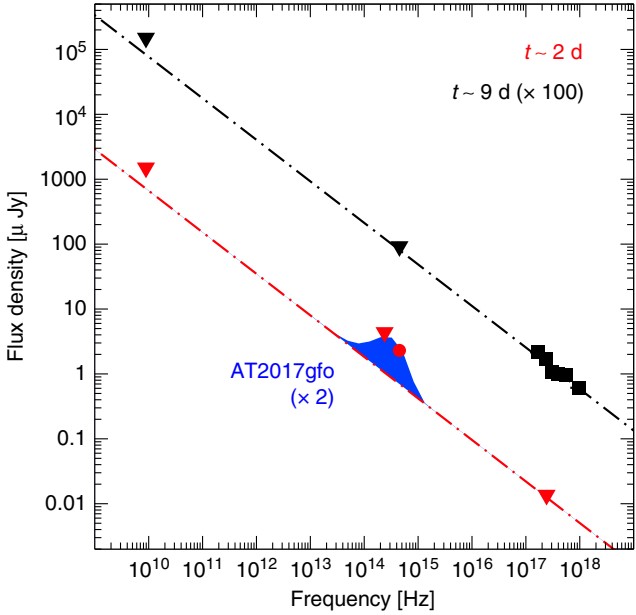

**Fig. 4** Spectral energy distribution of GRB150101B. Downward triangles are 3 σ upper limits. At early times, the optical light is brighter than expected based on the extrapolation of the X-ray constraints (dot-dashed line), suggesting that it belongs to a different component of emission. The blue area shows the spectrum of AT2017gfo[8,10,20] at 1.5 days, rescaled by a factor 2 in order to match the optical brightness of GRB150101B. At later times, the lack of optical detection is consistent with the X-ray afterglow behavior. At both epochs, our model is consistent with the lack of radio detection at 5.7 d[23]

At later times, this excess is no longer visible. The deep upper limit from *Gemini* shows that at ~10 days the afterglow became the dominant component and the kilonova already faded away. This is consistent with the behavior of AT2017gfo[8,10,20], and predicted in general by kilonova models[13,29,30].

**Environment**. Only a minor fraction (<30%) of short GRBs is associated to an early-type host galaxy[31]. This number decreases to ~18% if one considers only bona-fide associations, i.e., those with a low probability of being spurious. Notably, both GRB150101B and GW170817 were harbored in a luminous elliptical galaxy (Fig. 5).

We used the NASA/ESA *Hubble Space Telescope* (HST) to image the galaxy of GRB150101B (PI: Troja) in two filters, F606W and F160W, ~10 months after the burst. Earlier observations (PI: Tanvir) were performed ~40 days after the burst and are used to search for possible transient emission. Further photometric and spectroscopic studies were carried out with the 4.3 m Discovery Channel Telescope. Our results (Methods section) are in keeping with previous findings[23,32]. The galaxy morphology, its high luminosity (~4 $L^*$), old stellar population (~2 Gyr) and low on-going star formation rate (<~0.5 Msun yr$^{-1}$) closely resemble the properties of NGC 4993[33,34]. The location of GRB150101B is however farther off center than GW170817, 7.3 kpc away from the galaxy nucleus (~0.9 $r_{\rm eff;}$ Methods section). This is a region of faint optical and infrared light (Fig. 5), suggestive of a natal kick velocity for its progenitor, which merged far from its birth site.

**Discussion**
There is broad consensus that the properties of GW170817 can be explained by the emergence of a kilonova evolving from blue to

red colors[8,10,13,20], and a delayed afterglow component[8,11,12,21]. However, key aspects of this epochal event remain poorly understood. For instance, the luminous blue emission from AT2017gfo points to a large amount (~0.01 $M_{\rm sun}$) of rapidly expanding ($v \sim 0.2\ c$) ejecta with relatively low opacity[35,36], as expected if they are mainly composed of light r-process nuclei. This tail of massive, high-velocity ejecta is challenging to reproduce in standard NS mergers models[37,38], and its origin is still not clear. Another highly debated topic is the successful emergence of a relativistic jet, as the broadband afterglow data are well-explained either by a highly relativistic structured jet seen off-axis[8,12,14], or by a choked jet embedded in a mildly relativistic wide-angle cocoon[11,24]. In this context, the discovery of cases similar to GRB170817A, either through gamma-ray or GW triggers, is crucial to understand whether this burst was a peculiar event, a standard explosion or the prototype of a new class of transients.

The case of GRB150101B provides important information to help in this quest. This burst was located in an environment remarkably similar to GW170817, a luminous elliptical galaxy characterized by an old stellar population and no traces of on-going star formation. Therefore, despite the lack of a simultaneous GW detection, a NS merger origin for this burst is highly favored. The prompt phase of GRB150101B, although much shorter in duration, resembles in other aspects the phenomenology of GRB170817A[39]. Similar to GW170817, the weak gamma-ray emission of GRB150101B was accompanied by unusually bright optical and X-ray counterparts. As shown in Fig. 2a, b, this phenomenology does not match the typical behavior of short GRB afterglows and is suggestive of a GW170817-like explosion, in which the early-time optical emission is dominated by the kilonova and the X-ray emission traces the onset of a delayed afterglow.

If our interpretation is correct, GRB150101B is only the second known case of this phenomenological class of transients and its study can help us to delineate the general properties of these explosions, disentangling them from the particular properties of GW170817. First, the optical counterpart of GRB150101B is more luminous than AT2017gfo at the same epoch (by a factor of ~2), and fades at a slightly slower rate (~0.5 mag d$^{-1}$). Figure 4 shows that the afterglow contribution is sub-dominant at early times, and the most likely origin of the observed optical light seems a bright kilonova. Interestingly, its brightness and decay rate closely resemble the candidate kilonova from GRB050709[40]. By assuming an opacity $\kappa \sim 1\ {\rm g\ cm^{-3}}$ and imposing that the peak of the kilonova emission is $t_{\rm pk} < 2$ d we estimate a large mass >0.02 $M_{\rm sun}$ and velocity $v > 0.15c$ of lanthanide-poor ejecta (see Eq. 4 and 5 of ref. [41]), challenging to reproduce in standard simulations of NS mergers[37].

Our results suggest that a luminous blue kilonova might be a common feature of NS mergers rather than a peculiar characteristic inherent to GW170817, and reinforce the need for an efficient power source, such as a long-lived NS[42], to produce high optical luminosities at early times. While in the majority of previous short GRBs, searches were not sensitive to probe luminosities ~$10^{41}$ erg s$^{-1}$, in other cases the optical emission might have been misclassified as standard afterglow. Figure 6 compares the optical measurements of GRB150101B and other nearby short GRBs with the kilonova AT2017gfo. Whereas in some events, such as GRB050509B and GRB080905A, any kilonova component was either delayed or significantly fainter than AT2017gfo[43], in other cases the optical counterparts have luminosities comparable to AT2017gfo and could therefore be entirely powered by the kilonova emission or be a mixture of afterglow and kilonova. However, given the poor sampling and large errors of these datasets, it appears that past searches had the

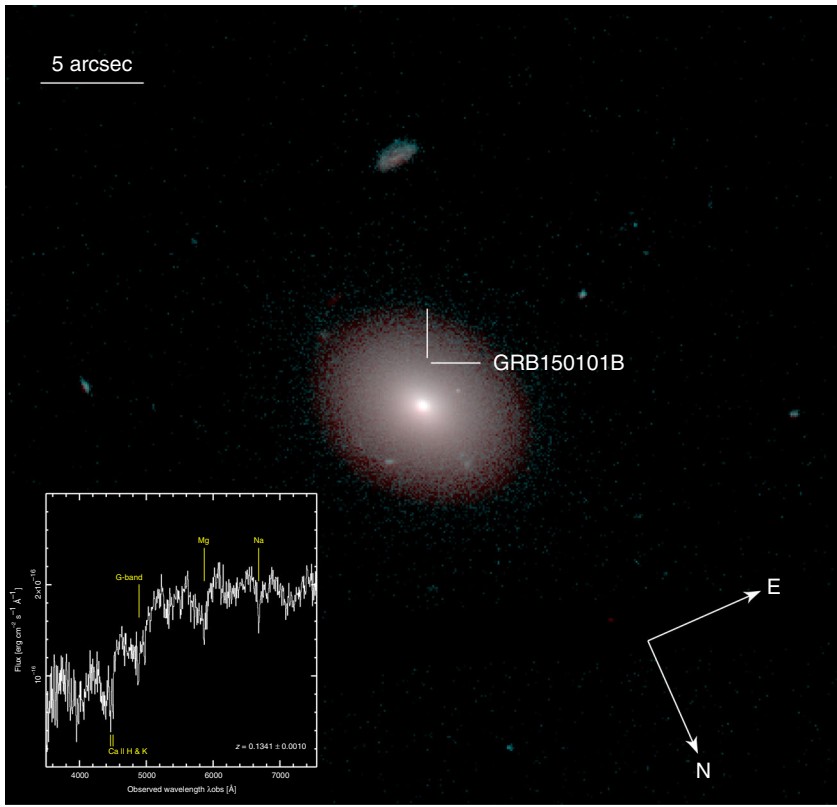

**Fig. 5** The host galaxy of GRB150101B. Color image of the host galaxy of GRB150101B created from the HST/WFC3 observations in filters F606W (blue, green) and F160W (red). The two intersecting lines mark the location of the GRB afterglow. The galaxy optical spectrum (inset) displays several absorption lines (Ca II H & K, G-band, Mg, and Na at $z = 0.1341$) and a red continuum

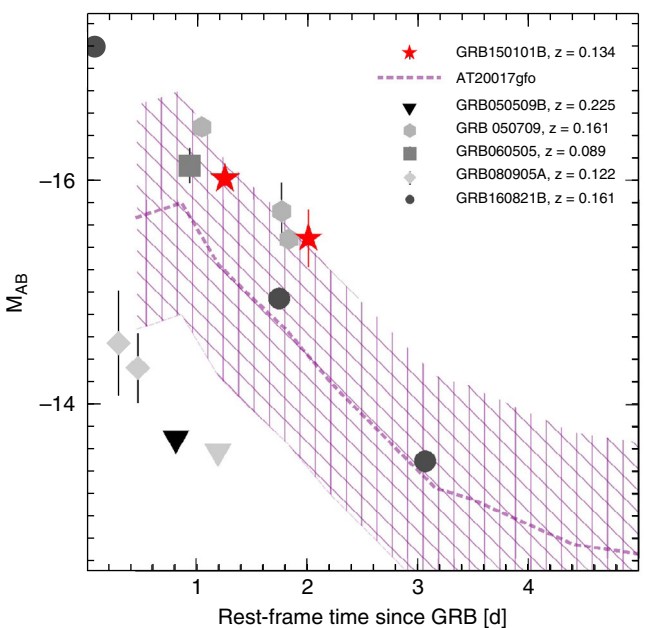

**Fig. 6** Early optical emission from short GRBs. Comparison between the kilonova AT2017gfo (dashed line; the hatched area shows the ±1 mag region) and the optical counterparts of nearby short GRBs. Vertical error bars are 1σ. Downward triangles are 3σ upper limits. Due to the limited spectral coverage of most events, we did not apply a k-correction and present the absolute magnitudes in the observed r-band filter. However, given the low redshift of these events, we expect this correction factor to be <0.5 mag. The data are from refs. [40,66–68]

sensitivity to detect these early blue kilonovae, but they were not adequate to identify their contribution. Rapid and deep multi-color imaging of future nearby short GRBs would be critical to probe the emergence of a kilonova component from the earliest times (~minutes after the merger) and disentangle it from any underlying afterglow. The optical to gamma-ray ratio, as presented in Fig. 2a, would also be a useful tool to weed out the most promising candidates. The presence of a luminous blue kilonova, if as common (although not ubiquitous) as suggested by our results, is promising for future follow-up observations of GW sources with wide-field optical monitors[44].

Unfortunately, the constraints at infrared wavelengths are not particularly sensitive to the emergence of a kilonova. The deep *HST* upper limit at 40 days can exclude some extreme scenarios with large masses (~0.03 $M_{sun}$) of lanthanide-rich ejecta. This limit is consistent with the range $M_{ej}$ ~0.01–0.1 $M_{sun}$ derived from the candidate kilonova in GRB130603B[45] and with the properties of the kilonova AT2017gfo, in which the red component only requires <0.001 $M_{sun}$ of high-opacity material, typical for a NS–NS merger.

At higher energies, the long-lived X-ray emission of GRB150101B and its low gamma-ray energy release resemble the behavior of GRB170817A and its late-peaking afterglow. We fit both the structured jet and cocoon models to the GRB150101B afterglow. Despite the high number of free parameters, the tight constraints of the X-ray and optical upper limits allow us to draw conclusions about both models (Table 1).

In the cocoon-dominated scenario[24,46,47], as the jet propagates through the merger ejecta, most of its energy is deposited into a hot cocoon, which then breaks out of the ejecta (while the jet is choked). The interaction of this mildly relativistic cocoon with the ambient medium produces a delayed afterglow. For

GRB150101B, the early peak of the X-ray emission implies a wide-angle cocoon with negligible energy injection and significant velocity ($\Gamma \sim 10$), difficult to reconcile with full spherical ejecta. The high X-ray luminosity at peak ($L_{X,pk} \sim 5 \times 10^{42}$ erg s$^{-1}$) implies that a high energy budget ($E_{k,iso} > 10^{52}$ erg) had to be transferred to the cocoon over a very short timescale ~0.01 s. This short duration of the gamma-ray emission is hard to explain within the cocoon model, as observational evidence[48] suggests that the breakout should typically occur on longer timescales of ~0.2–0.5 s.

In the structured jet scenario[8,12,14,49], the afterglow peak time $t_{pk}$ occurs when the central core of the jet decelerates sufficiently to come into view of the observer. It is a strong function of the viewing angle $\theta_v$ and, for $\theta_v < 0.7$ rad, occurs before any significant spreading takes place. For a Gaussian structured jet $t_{pk} \propto (\theta_v - \theta_c)^{2.5}$, where $\theta_c$ is the Gaussian width of the jet energy distribution. For GW170817, broadband afterglow modeling[26] robustly constrains $\theta_v \sim 25°$ and $\theta_c \sim 4°$. Since GRB150101B peaked ~10 times earlier than GW170817, it was likely seen at a smaller angle of $\theta_v - \theta_c \sim 10°$ for similar explosion properties. This is consistent with its higher value of $E_{\gamma,iso}$ and the short duration of its prompt gamma-ray phase. The peculiar properties of GRB150101B could therefore be due to the observer's orientation and, if observed on-axis, the burst would appear as a canonical short duration GRB.

We show that this hypothesis leads to a self-consistent scenario. We consider the $L_X/E_{\gamma,iso}$ diagram (Fig. 2b). The post-peak X-ray afterglow is not significantly affected by the viewing angle, and mostly resembles the properties (luminosity and decay rate) of an equivalent on-axis explosion. Since the post-peak $L_X$ is roughly the same for on-axis and off-axis jets, the observed $E_{\gamma,iso}$ can be used to probe the off-axis angle. In this particular case, the GRB would need an isotropic energy higher by a factor $E_{\gamma,on-axis}/E_{\gamma,obs} \sim 50$–500 for the observed $L_X/E_{\gamma,iso}$ ratio to fall within the 1 σ range of standard short bursts (shaded area in Fig. 2b). Indeed, this ratio would bring $E_{\gamma,on-axis}$ within the typical range of *Swift* bursts (Fig. 1). For a Gaussian shaped jet, the relation $\theta_v/\theta_c \sim (2\ln(E_{\gamma,on-axis}/E_{\gamma,obs}))^{1/2}$ must hold and implies a narrow core, $\theta_c \sim 3°$–5°, similar to GRB170817A and within the range observed in other short GRBs[50,51]. By using the models developed in ref. [12], we find that the luminosity and peak time of the X-ray light curve can be reproduced by an explosion with $E_k = 3\ (-2, +3) \times 10^{50}$ erg and $\theta_v = 13°\ (-5°, +7°)$ (Table 1). The jet model favors an environment of moderately low density, $n = 0.070\ (-0.069, +0.53)$ cm$^{-3}$, typical for bright elliptical galaxies[52]. The interaction between such medium and the relatively high mass of subrelativistic ejecta ($>0.02\ M_{sun}$) produced by the merger may give rise to a detectable radio signal on a timescale of a few years after the burst[53].

We conclude that the overall properties of GRB150101B are naturally accounted for by a structured jet viewed off-axis. This implies that in the nearby ($z < 0.2$) Universe, where off-axis explosions are detectable by current gamma-ray facilities[54], the true-to-observed ratio of short GRBs is much smaller than the beaming correction factor (~300) of the uniform jet model[51,55]. Based on our model of a Gaussian jet, we show in Fig. 7 some predictions for X-ray searches of GW counterparts. Future counterparts of GW sources might span a wide range of behaviors at X-ray energies, depending on the observer's orientation (in addition to the details of the explosion). For a fiducial distance of 100 Mpc, only a jet seen close to its axis ($\theta < 20°$), like GRB150101B, could be detected by the *Swift*/XRT. Due to the significant contribution of the jet wings to the emission, these afterglows may not display a clear temporal variability for weeks after the merger and require a long-term monitoring in order to be identified as transients. Explosions seen at larger viewing

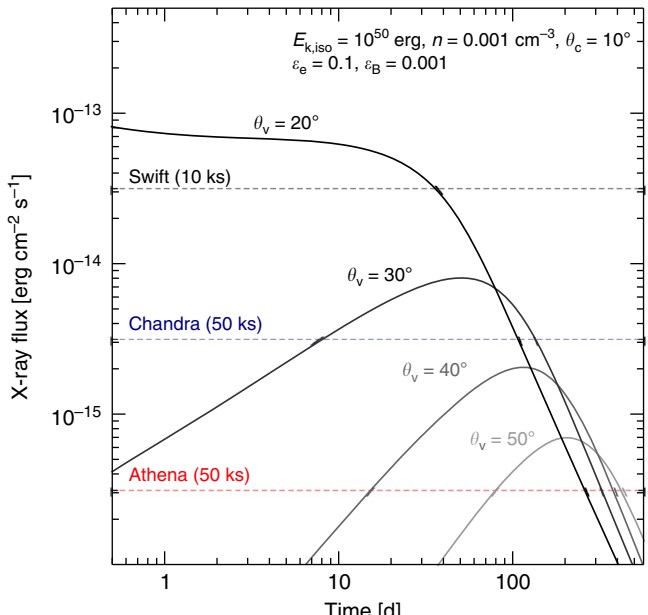

**Fig. 7** Predicted X-ray light curves of GW counterparts for different viewing angles. We assumed a set of standard afterglow parameters, reported in the top right corner, and a distance of 100 Mpc. The dashed lines mark the typical sensitivities of present (*Swift*, *Chandra*) and future (*Athena*) X-ray observatories

angles, such as GW170817, show a clear rising afterglow but require the sensitivities of *Chandra* and, in future, the *Athena X-ray Observatory* will be able to uncover a wider range of orphan X-ray afterglows.

## Methods

**Gamma-ray data reduction.** At 15:23:34.47 UT on 01 January 2015, GRB150101B triggered the Gamma-ray Burst Monitor (GBM) aboard NASA's *Fermi* satellite and was later found in a refined analysis of the *Swift* Burst Alert Telescope (BAT) data. These data were downloaded from the public *Swift* (https://heasarc.gsfc.nasa.gov/FTP/swift/data/obs/2015_01/) and *Fermi* (https://heasarc.gsfc.nasa.gov/FTP/fermi/data/gbm/triggers/2015/) archives, and analyzed using the standard tools within the HEASoft (version 6.23) package and the public version of the RMFIT software (https://fermi.gsfc.nasa.gov/ssc/data/analysis/rmfit/). Analysis of the untriggered INTEGRAL dataset also detects this event[56].

As seen by BAT, the GRB has a duration $T_{90} = 12 \pm 1$ ms in the 15–150 keV energy band. Search for emission on longer timescales did not detect any significant signal with a typical 3 σ upper limit of ~2.5 × 10$^{-7}$ erg cm$^{-2}$ s$^{-1}$ (15–50 keV). Due to the faintness of the burst, the BAT spectrum was binned into 10 energy channels rather than the standard 80 channels. The spectrum in the range 15–150 keV can be described by a simple power-law model with photon index $\Gamma = 1.2 \pm 0.3$ ($\chi^2 = 12$ for 8 d.o.f.) as well as a black body with k$T = 17 \pm 5$ keV ($\chi^2 = 9$ for 8 d.o.f.). Other non-thermal models, such as a power-law with an exponential cutoff, a log-parabola or a Band function, yield comparably good descriptions of the spectrum but have a higher number of free parameters. Given the faintness of this event, these results should be taken with a grain of salt. The blackbody fit suggests the presence of a spectral curvature above ~100 keV, not captured by the simple power-law model. It is however not possible to distinguish between thermal and non-thermal emission based on this dataset.

A comparison of the BAT and GBM datasets (Fig. 1) shows that the start of the GRB emission is delayed by a few milliseconds in BAT. The early prompt phase appears to peak at energies >350 keV, above the BAT bandpass, and is therefore not detected by *Swift*. We therefore added this hard component of emission to our spectral analysis in order to estimate the total burst energetics. The GBM spectrum was extracted from the three NaI detectors with the highest signal-to-noise (NaI$_6$, NaI$_7$, and NaI$_8$) and the relevant BGO$_1$ detector. The time-averaged spectrum is well described by a power-law with an exponential cutoff (C-STAT = 276 for 274 d.o.f.). The observed fluence in the 10–1000 keV energy band is $(1.0 \pm 0.2) \times 10^{-7}$ erg cm$^{-2}$. At a redshift $z = 0.1341$, this corresponds to an isotropic-equivalent luminosity $L_{\gamma,iso} = (2.7 \pm 0.6) \times 10^{50}$ erg s$^{-1}$ and total gamma-ray energy $E_{\gamma,iso} = (4.7 \pm 0.9) \times 10^{48}$ erg, calculated over the rest-frame 10–1000 keV band.

**Table 2 Observations of GRB150101B. Upper limits are 3 σ**

| MJD start | T−T0 [d] | Instrument | Band/filter | Exposure [s] | GRB counterpart flux[a] | Host flux[a] |
|---|---|---|---|---|---|---|
| *X-rays* | | | | | | |
| 57025.1 | 1.5 | *Swift*/XRT | 0.3–10 keV | 9880 | <1.5 | 3.7 |
| 57031.5 | 7.9 | *Chandra*/ACIS-S | 0.3–10 keV | 14,870 | 1.10 ± 0.13 | 3.71 ± 0.16 |
| 57063.2 | 39.6 | *Chandra*/ACIS-S | 0.3–10 keV | 14,860 | 0.16 ± 0.06 | 3.58 ± 0.16 |
| *Optical/nIR* | | | | | | |
| 57025.2[b] | 1.7 | *Magellan*/IMACS | r | 1200 | 23.01 ± 0.17 | — |
| 57026.2[b] | 2.7 | *Magellan*/IMACS | r | 1200 | 23.53 ± 0.26 | — |
| 57026.2 | 2.7 | *VLT*/HAWK-I | J | 720 | >22.5 | — |
| 57034.3 | 10.7 | *Gemini*/GMOS | r | 1710 | >24.2 | — |
| 57052.2 | 35.0 | *Swift*/UVOT | u | 1022 | — | 19.86 ± 0.04 |
| 57053.3 | 36.7 | *Swift*/UVOT | w2 | 398 | — | 21.30 ± 0.10 |
| 57064.3 | 40.7 | *HST*/WFC3 | F160W | 750 | >25.1 | — |
| 57064.3 | 40.7 | *HST*/WFC3 | F606W | 1750 | >25.6 | — |
| 57115.0 | 97.8 | *VLT*/HAWK-I | J | 2400 | — | 15.45 ± 0.02 |
| 57358.5 | 335 | *DCT*/DeVeny | 300 g/mm | 600 | — | Spectrum |
| 57372.1 | 348.5 | *HST*/WFC3 | F160W | 2400 | — | 15.14 ± 0.01 |
| 57372.2 | 348.6 | *HST*/WFC3 | F606W | 2500 | — | 16.67 ± 0.01 |
| 58258.0 | 1226 | *DCT*/LMI | g | 60 | — | 17.48 ± 0.01 |
| 58258.0 | 1226 | *DCT*/LMI | r | 60 | — | 16.50 ± 0.01 |
| 58258.0 | 1226 | *DCT*/LMI | i | 60 | — | 16.09 ± 0.01 |
| 58258.0 | 1226 | *DCT*/LMI | z | 60 | — | 15.80 ± 0.01 |

[a]X-ray fluxes are in units of $10^{-13}$ erg cm$^{-2}$ s$^{-1}$ and the observed-to-unabsorbed correction factor is -1.1. Optical and near-infrared magnitudes are in reported the AB system and corrected for Galactic extinction in the direction of the burst
[b]ref. [23]

These results differ from the preliminary analysis reported through GRB Circular Notices. These differences could be attributed, in part, to the poor signal to noise of the standard BAT spectrum used in that analysis. Our derived value of $E_{\gamma, iso}$ is lower than the value reported in ref. [23], which adopts an arbitrary k-correction factor of 5 to convert the fluence measured in the BAT bandpass to the 10–1000 keV range. In our case, such correction is not necessary as we measure the fluence over the broader energy bandpass. The fluence derived from the time-averaged spectral fit of the main burst is consistent with the value reported in ref. [39]. Their catalog[57,58] luminosities and energetics are instead calculated in a different energy range with a different methodology and cannot be directly compared to our values. The energetics for the entire sample of BAT short bursts with redshift are consistently derived from the time-averaged spectral properties following the same methodology used for GRB150101B.

**X-ray data reduction**. The data from the *Chandra* X-ray Telescope were reduced and analyzed in a standard fashion using CIAO v.4.9 and the latest calibration files. The afterglow counts were estimated from a circular aperture of 2-pixel radius in order to minimize the contamination from the nearby galaxy nucleus.

Spectra were fit within XSPEC with an absorbed power-law model by minimizing the Cash statistics. The final photometry is reported in Table 2. We used the tool *psfsize_srcs* to calculate that, within the selected source extraction region, the contamination of the central AGN is ~0.2% of its flux (~$7 \times 10^{-16}$ erg cm$^{-2}$ s$^{-1}$) and can be considered negligible in both epochs.

*Swift* XRT data were retrieved from the public repository (http://www.swift.ac.uk/xrt_curves/). Due to the larger point spread function (PSF) of the XRT, the two nearby sources resolved by *Chandra* are blended into a single source in the XRT images. Nonetheless these data can still be used to constrain the afterglow brightness. We focused on the first XRT observation at 1.5 d which resulted in a total of 115 source counts in a 9.9 ks exposure. We folded the AGN spectrum through the XRT response function in order to constrain the count rate of the central source. The XRT image was then modeled using two PSF-shaped sources at an offset fixed to the value measured with *Chandra*. This method allows us to place a 3 σ upper limit of $1.5 \times 10^{-13}$ erg cm$^{-2}$ s$^{-1}$ on the afterglow flux at 1.5 d.

**Ultraviolet optical and infrared data reduction**. We analyzed the observations from the 4.3 m discovery channel telescope (DCT), the 8.1 m Gemini-South telescope, and the 8.2 m ESO's very large telescope (VLT). The data were reduced within IRAF[59] using standard tools and CCD reduction techniques (e.g., bias subtraction, flat-fielding, etc.). At later times, the location of the burst was imaged with the NASA/ESA hubble space telescope (HST) at two epochs, 40 days and 10 months after the burst. The later observation was used as a template for image subtraction. In each epoch, the optical F606W filter (0.6 μm) and the near-infrared F160W filter (1.6 μm) were used. Upper limits on the source flux were calculated by

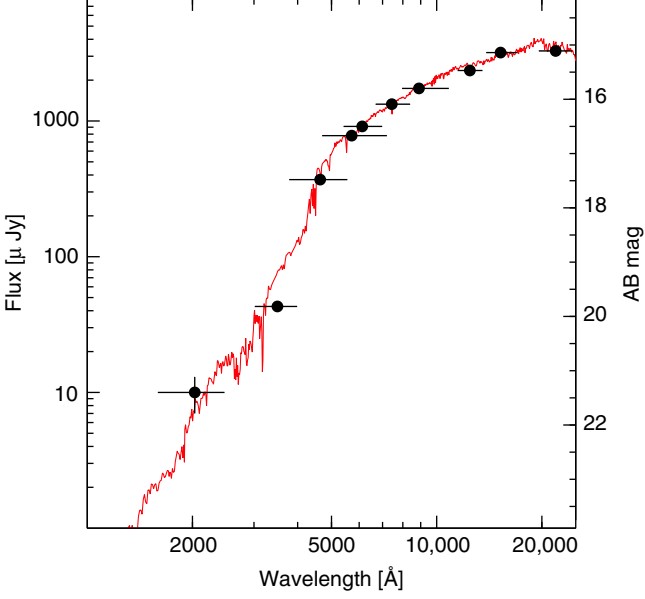

**Fig. 8** Spectral energy distribution of the GRB host galaxy. The best fit model is shown by the solid line

placing artificial sources of known magnitude at the transient location and recovering their fluxes through PSF-fitting on the subtracted image.

Observations with the ultraviolet and optica telescope (UVOT) aboard *Swift* were reduced and analyzed using HEASoft v6.23. Count-rates were estimated from a 5 arcsec aperture region and converted into magnitudes using the UVOT zero points[60]. The final photometry is reported in Table 2.

**Host galaxy**. We used the late-time HST images to model the surface brightness (SB) profile of the host galaxy using the procedure described in ref. [33]. Stars in the vicinity of the galaxy were used to construct an empirical PSF used as input to GALFIT[61] for the two-dimensional SB fitting. We tried several SB models, including a single Sèrsic profile, or a double Sèrsic profile ($n = 4$ for the bulge and $n = 1$ for the disk), with or without an additional point source at center. We find

that the SB profile of the host galaxy is best fitted when the point source component is included in both bands. The need for the central point source component suggests that this galaxy harbors a low luminosity AGN with F606W ~20.8 AB mag, and F160W ~18.2 AB mag, taking up to 2% and 6% of the total light in each band, respectively.

Both the single and double Sèrsic fit results suggest that this is an early-type galaxy, with a Sèrsic index of $n = 5.1 \pm 1.0$ (F606W) or $n = 2.4 \pm 1.0$ and B/T ~ 0.7–0.8. Morphological appearance of this object also supports this result. The effective radius, $r_{eff}$ is found to be 3.4 arcsec (or 8.1 kpc at $z = 0.1341$) to 3.9 arcsec (9.3 kpc) in F606W, and ~2 arcsec (6 kpc) in F160W. The position of GRB150101B was found to be 3 arcsec (~7.3 kpc) from the galaxy center, meaning that the GRB occurred around $r_{eff}$ of the host galaxy.

The properties of the stellar population were constrained by modeling the galaxy spectral energy distribution. We excluded the AGN component assuming a composite AGN SED[62,63] and normalizing the AGN flux at the value measured in the F160W filter. We fixed the redshift to 0.1341 and used the Chabrier initial mass function[64]. Two SED fitting procedures were tried, that of ref.[64] and the fitting and assessment of synthetic templates[65] (FAST). Both fitting methods return consistent results (Fig. 8): a best-fit mean stellar age of 2 (+6, −1) Gyr, stellar mass of 1.0 (+1.0, −0.2) × $10^{11}$ $M_{sun}$, and star formation rate SFR ~0.5 $M_{sun}$ yr$^{-1}$.

## Data availability

The data that support the findings of this study are available from the corresponding author upon reasonable request.

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

## Acknowledgements

We thank B. Wilkes and the *Chandra* staff for approving and rapidly scheduling our observations under Director's Discretionary Time. We thank A. Y. Lien for help with the *Swift* BAT data, P. A. Evans for suggestions on the *Swift* XRT analysis, and P. D'Avanzo for providing the VLT/HAWK-I images. E.T. acknowledges the contribution of Bianca A. Vekstein to the *Chandra* proposal. These results made use of the Discovery Channel Telescope at Lowell Observatory. Lowell is a private, non-profit institution dedicated to astrophysical research and public appreciation of astronomy and operates the DCT in partnership with Boston University, the University of Maryland, the University of Toledo, Northern Arizona University, and Yale University. The Large Monolithic Imager was built by Lowell Observatory using funds provided by the National Science Foundation (AST-1005313). The upgrade of the DeVeny optical spectrograph has been funded by a generous grant from John and Ginger Giovale. Support for this work was partially provided by the National Aeronautics and Space Administration through grants HST GO-13941, 14087, and 14357 from the Space Telescope Science Institute. M.I., Y.Y., and S.-K.L. acknowledge support from the National Research Foundation of Korea (NRF) grant, No. 2017R1A3As3001362. Analysis was performed on the YORP cluster administered by the Center for Theory and Computation, part of the Department of Astronomy at the University of Maryland.

## Author contributions

E.T. led the data analysis and paper writing with inputs from L.P., G.R., S.B.C., H.v.E. and T.S. G.R. and H.v.E. developed the afterglow models and performed the afterglow fits. Y. Y., S.-K.L. and M.I. derived the properties of the host galaxy. S.B.C., P.G., A.K. and S.V. contributed to the acquisition and data reduction of the DCT data.

## Additional information

**Competing interests:** The authors declare no competing interests.

