## [Peer Review File · Nature Communications]

Reviewers' comments:

Reviewer #1 (Remarks to the Author):

The manuscript by Troja et al. reports that a faint short GRB at a cosmological distance (GRB150101B) and its late time emission are analogous to the neutron star merger event GRB 170817A. It is shown that their properties are consistent with off-axis jet models, and these could be common features for electromagnetic (EM) counterparts of GW sources. The results presented in this paper are interesting and useful for EM counterpart study. Considering that the subject is very hot, the paper is like to trigger further discussion in the field. It certainly deserves to be published in Nature Communications.

My remaining concern is their X-ray analysis. Their interpretation heavily relies on the decaying behavior of the faint x-ray source observed by Chandra (7.9d and 39.6d after the burst). The X-ray data reduction section in "Methods" should be expanded to clearly indicate the estimates of the contamination from the nearby galaxy nucleus.

Early afterglow often shows a complicated behaviour (flares, bumps, chromatic breaks). Although we do not understand yet what causes it, certainly kilonovae are not a favored explanation in most cases. The additional optical component at 2d indicated in Figure 4 can be due to a different process, rather than a kilonova. Is the colour unusual for bumps or flares observed in other short/long events? Explain why a kilonova is a favored explanation for this event.

typo: x-axis label in Fig3 Time [s] → Time [d].

Reviewer #2 (Remarks to the Author):

This is a report on "A luminous blue kilonova and an off-axis jet from a compact binary 1 merger at $z=0.1341$ " by Troja et al. (manuscript number NCOMMS-18-19107).

The manuscript describes an event with properties similar to those of the electromagnetic emission associated with a binary neutron star merger GW170817: an underluminous short gamma ray burst, an optical kilonova, and a broad-band afterglow that rises on relatively long timescales.

Finding another such event, GRB150101B, in archival data establishes the existence of a class of similar events. This has significant implications for our understanding of compact binary mergers, outflow geometries, and kilonova engines. Therefore, the paper is an important contribution to the field, and I would in principle recommend that it be published in Nature Communications if the following issues can be addressed.

My biggest concern is that the data used to support the claim that GRB150101B belongs in the same class of events as GW170817 are presented in a contradictory way in different figures and tables. For example, figure 2A shows two different optical detections (at <2 days and around 3 days) supporting the kilonova hypothesis. Figure 4, on the other hand, suggests that there is only one optical observation at ~ 2 days, and one upper limit. Meanwhile, table II lists one optical/nIR upper limit at 2.7 days, and no optical detections of the transient at all! So which is it: are there 0, 1, or 2 optical detections at ~ 2 days? There are similar issues with other observations. The authors should list **all** of the data behind the figures in table II, and ensure that all figures are fully consistent with these data.

The second issue is one of overall text and figure editing and presentation. While the discussion section is generally clear, the earlier sections are frequently less so, with some material not

adequately explained until later, and other points repeated multiple times with varying levels of detail. I recommend moving some of the discussion material earlier in the paper to present a clear framework for the results and avoid unnecessary repetition.

Other issues follow in order of the paper:

line 82: $T_{90} \sim 12$ ms -- how does that compare with GRB170817A?

line 89: follow-up*s* of short GRBs

Figure 1: the axis labels are almost illegible without magnification. How can counts per second be negative -- is there some mean background being subtracted or other processing? The error bars appear to be inconsistent with Poisson statistics -- how were they obtained? The values of a few counts per second appear to be integrated over ~ 0.001 s bins -- so how are there any counts in such bins?

Figure 2: the axis labels are almost illegible without magnification.

The vertical axes labeled as L/E; since this is a dimensional quantity, it should have units. AT2017gfo is used in the left panel, GW170818 is used in the right panel, GRB170817A is used in the caption... I could not figure out the rationale for determining which of the labels is used here or in the text; at the very least, it would be useful to the reader to say that all three refer to the same event at the very beginning of the paper.

The red "afterglow" curve in the left panel is an extrapolated model (there are only upper limits at this time), so should be more clearly labeled as such (or, better, the curve should be omitted).

I could not find the 1 sigma error bars promised in the caption.

The dotted and dashed lines used to describe the kilonova and afterglow were difficult to distinguish -- the right panel appears to show a kilonova, though it actually shows the afterglow. Why not show the optical afterglow data from Lyman et al., 2018, for GW170817 in the left panel?

Line 159: 0.0015 cts/s is very different from what is shown in figure 1!

Figure 3: The horizontal axis should presumably have units of days, not seconds.

The dashed black line in the top panel is not a power law... ahh, OK, it is the superposition of a power law with a constant background -- this could be explained more clearly.

How are meaningful 1-sigma error bars computed on a model which is clearly a terrible fit to the data?

Why does the very broad cocoon rise much faster than the narrower structured jet at early times?

Line 205: spectral index beta: provide the expression defining beta

Line 206: electron slope p: electrons do not have slopes; proper terminology should be used

Line 209: how is 1.5×10^{-13} erg / cm² / s obtained? "upper limit *on*" "the same spectral shape *as*"

Figure 4 and associated discussion: the use of a single spectral shape throughout is not obvious, given the limited data (and the apparent contradiction, which should be discussed, between the inferred spectrum and the 3-sigma optical limit at 9 days). E.g., could the cooling frequency shift relative to the observing frequency?

Line 246: mention the 7.3 kpc distance (currently only mentioned in figure 5 caption) in the main text; compare with host galaxy effective radius

Figure 5: "prominent break downward 4000 angstroms" -- where is this prominent break on the

inset? [aside: inset labels virtually illegible]

Line 286: "fades at a slightly lower rate" -- so what is this rate (and what are the error bars, if it is established from ≤ 2 data points)?

Line 289: $\kappa \sim 1 \text{ g / cm}^3$ -- since the kilonova appears blue, this would argue for fewer high-opacity elements in the ejecta, and a lower opacity would yield an even higher mass?

Line 284: why is radioactive-powered emission insufficient? If the only conflict is in the models for ejected mass, how much room for uncertainty is there in those models? "need *for* an efficient power source"

Line 303: "These data set*s*"

Figure 6: do downward pointing triangles mean upper limits as previously (they aren't listed in the legend)? The figure shades and legend shades did not quite match, so was challenging to match the two.

Lines 336--340: not clear where some of the claimed values such as $\Gamma \sim 10$ and $E_{k,iso} > 10^{52} \text{ erg}$ came from; are these based on fitting particular models, and if so, what other parameters (e.g., ISM density) do these depend on?

Line 343: peak time scaling with observing angle depends on assumptions about sideways spreading of jet; clarify these.

Line 344: is it assumed here that the structured jet angular scale θ_c is universal? [Calling this the "core half-opening angle" is rather misleading for a Gaussian structured jet, which doesn't have a clearly defined core boundary.]

Line 350: "In order to test this hypothesis" -- it is hardly testing a hypothesis when the number of free model parameters is larger than the the number of observations! The authors show that a particular structured jet model has the freedom to match the sparse data.

Lines 360, 361: error bars (which are very sizeable -- see table 1) should be given on inferred quantities such as E_k and n [e.g., if one-sigma error bars on n are between 0.001 cm^{-3} and 0.6 cm^{-3} , quoting it as 0.07 cm^{-3} without any ranges is rather misleading].

Line 364: if the X-ray afterglow started to decay 10 days after the burst, and the radio afterglow follows the same spectrum at all times, why would the source become visible a few years after the event (i.e., now?!) in the radio?

Line 365: "true-to-observed ratio of short GRBs is much smaller" -- is this actually true, since, as the authors point out, off-axis short GRBs are very underluminous in their model and are therefore unlikely to be detectable except in the relatively local Universe?

Line 368: "model of *a* *G* gaussian jet"

Table 1: would be helpful to remind readers what these parameters mean (since definitions are spread throughout the text); e.g., what is E_0 (as opposed to E_k and $E_{k,iso}$ that are used in the text, and also not always clearly defined)?

Non-standard normalisations for some of the variables (e.g., n) require the reader to do unnecessary mental arithmetic

Finally, I would have liked to see some discussion of whether there may be other events like this lurking in the archival data, or whether this was truly exceptional because of the follow-up

campaign. Even if this event is exceptional, is it possible to make statistical inferences about other apparently underluminous short GRBs? Are there specific strategies to follow these up even in the absence of gravitational-wave signals?

Reviewer #3 (Remarks to the Author):

Dear Authors,

I have read with great pleasure your manuscript and I think it is well written, clear, and convincing in its conclusions. Definitely to be published.

I do not have really major comments, but there are moderately important issues that I think should anyway be addressed to further improve the paper.

. Since the submission of the paper we have now an improved view of the late evolution of the GRB170817A (e.g. Lyman et al. 2018, arXiv:1801.02669). Although already now the discussion is fair, I would suggest to update the paper according to these new findings when you refer to GW170817.

. In the literature, hints of kilonova detections were already reported. I refer here at least to the papers by Tanvir et al. (2013, Nature 500, 547), Yang et al. (2015, NatComm 7, 7323), Jin et al. (2015, ApJ 811, 22) and Jin et al. (2016, NatComm 7, 12898). You mention and discuss already the last one. Now, I am well aware of the limitations on the number of references, and which papers should be quoted is entirely a choice of the authors. However, I warmly suggest to better stress that, in spite of the fact that your analysis is of great interest and meaning, there have been remarkable previous efforts to identify elusive kilonova signatures in short GRB light-curves.

. One thing that appears immediately clear to the reader is the strongly different result obtained in this paper compared to the Fong et al. (2016) paper. There is nothing bad in it, however you should discuss and summarise, briefly, the main reasons for that.

Then a few minor comments, often just typos:

. End of the Introduction. It could be interesting to also quote the percentage of Swift short GRBs with no accurate localisation.

. 84. I do not know if you are using latex, yet it could be better to write “ $\{\rm iso\}$ ”.

. 106. A factor 2 brighter?, 2 times brighter?

. Temporal analysis section. Here, if required (only if required), you could save some space with the discussion of the AGN analysis that could be shorter.

. 226. Could you quantify how much brighter?

. 362. How does the derived density compare with what typically obtained analyzing short and long GRB afterglows?

. Gamma-ray data reduction Methods section. I think that the concept of “statically favored” fit is often ambiguous. There are, given a confidence level, acceptable or not acceptable fits. Having the smallest residuals is not enough to claim, in general, that a fit is better than another. Your

following discussion is indeed correct, and therefore I would just suggest to slightly re-formulate the paragraph.

. 464. 2-pixel radius?

Reviewers' comments:

Reviewer #1 (Remarks to the Author):

The manuscript by Troja et al. reports that a faint short GRB at a cosmological distance (GRB150101B) and its late time emission are analogous to the neutron star merger event GRB 170817A. It is shown that their properties are consistent with off-axis jet models, and these could be common features for electromagnetic (EM) counterparts of GW sources. The results presented in this paper are interesting and useful for EM counterpart study. Considering that the subject is very hot, the paper is like to trigger further discussion in the field. It certainly deserves to be published in Nature Communications.

My remaining concern is their X-ray analysis. Their interpretation heavily relies on the decaying behavior of the faint x-ray source observed by Chandra (7.9d and 39.6d after the burst). The X-ray data reduction section in “Methods” should be expanded to clearly indicate the estimates of the contamination from the nearby galaxy nucleus.

We used the tool *psfszsize_srcs* to calculate that, within the source extraction region, the expected contamination from the central AGN is ~0.2% of its flux ($\sim 7 \times 10^{-16}$ erg cm⁻² s⁻¹). This is >10 times fainter than the afterglow flux and can be considered negligible in both epochs.

This information was added to the “Methods” section as requested by the referee.

Early afterglow often shows a complicated behaviour (flares, bumps, chromatic breaks). Although we do not understand yet what causes it, certainly kilonovae are not a favored explanation in most cases. The additional optical component at 2d indicated in Figure 4 can be due to a different process, rather than a kilonova. Is the colour unusual for bumps or flares observed in other short/long events? Explain why a kilonova is a favored explanation for this event.

Given the limited data set, we cannot exclude that the optical excess is due to an odd behavior of the afterglow (e.g. flares). However, these types of chromatic features are usually observed at X-ray rather than at optical wavelengths, typically occur within a few hours after the GRB, and are more frequent in long GRBs than in short GRBs (e.g. Chincarini et al. 2007, Swenson et al. 2013). The luminosity and timescales of the observed optical excess more naturally fit within the kilonova scenario, as shown in Figure 6.

We added a paragraph to the main text to discuss this point.

typo: x-axis label in Fig3 Time [s] —> Time [d].

Fixed.

Reviewer #2 (Remarks to the Author):

This is a report on "A luminous blue kilonova and an off-axis jet from a compact binary 1 merger at $z=0.1341$ " by Troja et al. (manuscript number NCOMMS-18-19107).

The manuscript describes an event with properties similar to those of the electromagnetic emission associated with a binary neutron star merger GW170817: an underluminous short gamma ray burst, an optical kilonova, and a broad-band afterglow that rises on relatively long timescales.

Finding another such event, GRB150101B, in archival data establishes the existence of a class of similar events. This has significant implications for our understanding of compact binary mergers, outflow geometries, and kilonova engines. Therefore, the paper is an important contribution to the field, and I would in principle recommend that it be published in Nature Communications if the following issues can be addressed.

My biggest concern is that the data used to support the claim that GRB150101B belongs in the same class of events as GW170817 are presented in a contradictory way in different figures and tables. For example, figure 2A shows two different optical detections (at <2 days and around 3 days) supporting the kilonova hypothesis. Figure 4, on the other hand, suggests that there is only one optical observation at ~ 2 days, and one upper limit. Meanwhile, table II lists one optical/nIR upper limit at 2.7 days, and no optical detections of the transient at all! So which is it: are there 0, 1, or 2 optical detections at ~ 2 days? There are similar issues with other observations. The authors should list **all of the data behind the figures in table II, and ensure that all figures are fully consistent with these data.**

We agree with the referee that the data reported in the table were not consistent with the ones presented in the figures and apologize for the confusion. We think much of the confusion arose because we skipped the two detections at 1.5 d and 2.5 d, and only reported the upper limits. This is now fixed in the revised version.

The second issue is one of overall text and figure editing and presentation. While the discussion section is generally clear, the earlier sections are frequently less so, with some material not adequately explained until later, and other points repeated multiple times with varying levels of detail. I recommend moving some of the discussion material earlier in the paper to present a clear framework for the results and avoid unnecessary repetition.

We tried to improve the clarity of the Results section. In particular, we now introduce the jet and cocoon models in the section *Temporal Analysis* rather than in the discussion. Some points, which we deemed important, are intentionally repeated so that, even if a reader skips the results sections, he/she can still understand the content of the discussion.

Other issues follow in order of the paper:

line 82: T₉₀ ~ 12 ms -- how does that compare with GRB170817A?

This is 40 times shorter than GRB170817A. However, the duration of GRB170817A is based on the observations of another instrument in a different energy range, and a one-to-one comparison of the two numbers is not accurate.

line 89: follow-up*s* of short GRBs

we fixed it as “follow-up observations”

Figure 1: the axis labels are almost illegible without magnification. How can counts per second be negative -- is there some mean background being subtracted or other processing? The error bars appear to be inconsistent with Poisson statistics -- how were they obtained? The values of a few counts per second appear to be integrated over ~0.001 s bins -- so how are there any counts in such bins?

We are following the journal guidelines on font size and format for figures. We enlarged all the figures in order to make their labels easier to read.

The gamma-ray light curve is derived from *Swift*/BAT and is mask-weighted. The mask-weighting technique removes the background contribution. We added this information to the caption. Due to the mask-weighting process, the error bars follow a Gaussian distribution and are automatically calculated by the *Swift* tools used in the analysis.

The time bin is 2 ms. The units of the y-axis are counts/sec/detector and are standard units to report *Swift* BAT light curves. They cannot easily be converted into counts/sec. By inspecting the raw light curve (i.e. before the standard mask-weighting processing), we estimate that there are ~30 source counts in a 2 ms time bin at peak. We prefer to retain the units of counts/sec/detector as they are the standard BAT units and allow for an immediate comparison between this light curve and other GRB light curves seen by BAT.

Figure 2: the axis labels are almost illegible without magnification. The vertical axes labeled as L/E; since this is a dimensional quantity, it should have units. AT2017gfo is used in the left panel, GW170818 is used in the right panel, GRB170817A is used in the caption... I could not figure out the rationale for determining which of the labels is used here or in the text; at the very least, it would be useful to the reader to say that all three refer to the same event at the very beginning of the paper.

We added the units to the plot. We also clarified at the beginning of the paper that we generically refer to this event as GW170817, and specifically refer to AT2017gfo when discussing the kilonova emission, whereas we use GRB170817A when discussing the GRB and afterglow emission. As suggested by the referee we clarify that all these nomenclatures refer to the same event.

The red "afterglow" curve in the left panel is an extrapolated model (there are only upper limits at this time), so should be more clearly labeled as such (or, better, the curve should be omitted).

The curve is derived from the fit of the X-ray data and extrapolated to the optical band. We clarified it in the figure legend. We think it is important to show that the afterglow component is consistent with the optical upper limits.

I could not find the 1 sigma error bars promised in the caption.

Fixed. In some cases the error bars are smaller than the symbol size and are still not visible.

The dotted and dashed lines used to describe the kilonova and afterglow were difficult to distinguish -- the right panel appears to show a kilonova, though it actually shows the afterglow.

We improved the clarity of this plot in the revised version.

Why not show the optical afterglow data from Lyman et al., 2018, for GW170817 in the left panel?

We tried to include the late-time afterglow of GW170817 but the overall result did not seem optimal. The GW optical afterglow becomes visible and peaks at ~100-150 days, which would require us to excessively squeeze the x-axis. In the caption we mention that the optical afterglow becomes visible at later times.

Line 159: 0.0015 cts/s is very different from what is shown in figure 1!

The quoted count rate refers to the X-ray afterglow level, shown in Figure 2. However, since Figure 2 reports the flux and not the count rate, we removed the count rate level in order to avoid confusion.

Figure 3: The horizontal axis should presumably have units of days, not seconds.

Fixed

The dashed black line in the top panel is not a power law... ahh, OK, it is the superposition of a power law with a constant background -- this could be explained more clearly.

We clarified it in the caption.

How are meaningful 1-sigma error bars computed on a model which is clearly a terrible fit to the data?

We removed the model error regions from the figure.

Why does the very broad cocoon rise much faster than the narrower structured jet at early times?

The cocoon model (with negligible energy injection) is indeed characterized by a sharp rise of the emission, similar to the standard top-hat jet seen off axis (see for example Figure 3 of Hallinan et al. 2017). At early times, the cocoon is in the coasting phase and rises like t^3 . Deceleration begins when the light curve peaks at ~7 days.

In the structured jet scenario, deceleration already started for the slower moving material from the wings (see Figure 3 of Troja et al. 2018).

Line 205: spectral index beta: provide the expression defining beta

Added.

Line 206: electron slope p: electrons do not have slopes; proper terminology should be used

We reworded this sentence.

Line 209: how is 1.5×10^{-13} erg / cm² / s obtained? "upper limit *on*" "the same spectral shape *as*"

The methodology used to derive the upper limit is described in the Methods section on the X-ray analysis.

Figure 4 and associated discussion: the use of a single spectral shape throughout is not obvious, given the limited data (and the apparent contradiction, which should be discussed, between the inferred spectrum and the 3-sigma optical limit at 9 days). E.g., could the cooling frequency shift relative to the observing frequency?

The spectral slope derived from the first *Chandra* observation suggests that the cooling frequency at 7.8 d is still above the X-ray band, and therefore it does not affect the earlier afterglow behavior.

The cooling frequency slowly evolves with time as $t^{-0.5}$. However, its temporal evolution changes after the jet-break and basically remains constant. Therefore, in the jet model, the cooling frequency should stop its decay soon after the afterglow peak at ~10-15 days. This is consistent with our assumption of no visible spectral evolution up to ~40 d.

We do not claim that we can constrain the spectral shape below the optical range. The spectrum could peak above the radio frequencies and, in this case, the consistency with the radio upper limits would be trivial. The single spectral shape is the scenario which predicts the highest flux and Figure 4 shows that, even in this case, we are consistent with the entire broadband dataset. We also note that this assumption is observationally supported by the broadband afterglow of GW170817, which shows a single spectral shape from 9 d to 250 d after the merger.

We do not find any significant contradiction between the inferred spectrum and the optical upper limit at 9 d, which lies above the extrapolation of the X-ray spectrum in Figure 4.

Line 246: mention the 7.3 kpc distance (currently only mentioned in figure 5 caption) in the main text; compare with host galaxy effective radius

We added this to the main text. The effective radius is derived in the Methods section.

Figure 5: "prominent break downward 4000 angstroms" -- where is this prominent break on the inset? [aside: inset labels virtually illegible]

We removed this sentence as it did not refer to the spectrum shown in this figure.

Line 286: "fades at a slightly lower rate" -- so what is this rate (and what are the error bars, if it is established from ≤ 2 data points)?

The optical light fades at a rate of $0.5 \pm 0.3 \text{ mag d}^{-1}$. We added this information to the text.

Line 289: $\kappa \sim 1 \text{ g / cm}^3$ -- since the kilonova appears blue, this would argue for fewer high-opacity elements in the ejecta, and a lower opacity would yield an even higher mass?

An opacity of $\sim 1 \text{ g / cm}^3$ is a factor of ~ 10 lower than the opacity of lanthanide-rich ejecta, and we expect it to be a reasonable estimate for the opacity of a lanthanide-poor component. A lower opacity would indeed require a higher ejecta mass.

Line 284: why is radioactive-powered emission insufficient? If the only conflict is in the models for ejected mass, how much room for uncertainty is there in those models?

A radioactive-powered kilonova can achieve the observed luminosity and early timescales only for a large mass of ejecta moving at sub-relativistic velocity. Numerical simulations of mergers cannot reproduce such large quantity of ejecta moving so fast. The masses derived from kilonova observations are a factor of ~ 3 higher than the maximum mass of ejecta derived from simulations.

"need *for* an efficient power source"

Fixed.

Line 303: "These data set*s*"

Fixed.

Figure 6: do downward pointing triangles mean upper limits as previously (they aren't listed in the legend)? The figure shades and legend shades did not quite match, so was challenging to match the two.

We fixed the legend shades and apologize for the confusion. We explained in the caption that downward triangles are upper-limits.

Lines 336--340: not clear where some of the claimed values such as $\Gamma \sim 10$ and $E_{k,iso} > 10^{52} \text{ erg}$ came from; are these based on fitting particular models, and if so, what other parameters (e.g., ISM density) do these depend on?

These values were derived from the afterglow fit reported in Table 1. We added some text in the discussion to clarify this point.

Line 343: peak time scaling with observing angle depends on assumptions about sideways spreading of jet; clarify these.

We added some text to further clarify the peak time scalings.

Line 344: is it assumed here that the structured jet angular scale θ_c is universal? [Calling this the "core half-opening angle" is rather misleading for a Gaussian structured jet, which doesn't have a clearly defined core boundary.]

We clearly define θ_c as the width of the Gaussian energy distribution.

Line 350: "In order to test this hypothesis" -- it is hardly testing a hypothesis when the number of free model parameters is larger than the the number of observations! The authors show that a particular structured jet model has the freedom to match the sparse data.

We reworded this sentence. However, we note that the requirements on the afterglow peak time, $t_{pk} \sim 10-15$ d, and on the gamma-ray energy, whose on-axis value should fall within the typical range of short GRBs, are quite constraining for the jet and viewing angles. Despite the sparse data set, these two quantities are quite well constrained and, in general, the jet model provides a self-consistent description of the broadband emission for a range of typical afterglow parameters. As a comparison, the cocoon model can also reproduce well the afterglow data but the resulting parameters are difficult to explain. In particular, it requires quite a large amount of energy to be dumped into the system on very short timescales.

Lines 360, 361: error bars (which are very sizeable -- see table 1) should be given on inferred quantities such as E_k and n [e.g., if one-sigma error bars on n are between 0.001 cm^{-3} and 0.6 cm^{-3} , quoting it as 0.07 cm^{-3} without any ranges is rather misleading].

We added the errors to the text.

Line 364: if the X-ray afterglow started to decay 10 days after the burst, and the radio afterglow follows the same spectrum at all times, why would the source become visible a few years after the event (i.e., now?!) in the radio?

The afterglow emission is due to the interaction between the relativistic jet and the surrounding medium, and we agree that the radio afterglow should follow the same rapid decay of the X-ray emission. However, the interaction between the sub-relativistic ejecta and the surrounding medium also gives rise to a radio signal on timescales of a few years after the merger. This is a component of emission different from the afterglow. We clarify in the text that we refer to the effects of these sub-relativistic ejecta.

Line 365: "true-to-observed ratio of short GRBs is much smaller" -- is this actually true, since, as the authors point out, off-axis short GRBs are very underluminous in their model and are therefore unlikely to be detectable except in the relatively local Universe?

We clarify that this is valid in the nearby Universe ($z < 0.2$), where such events are detectable by our gamma-ray facilities. This becomes relevant when trying to derive the rate of events from the GBM sample, which presumably contains a higher number of this local off-axis events.

Line 368: "model of a Gaussian jet"

Fixed.

Table 1: would be helpful to remind readers what these parameters mean (since definitions are spread throughout the text); e.g., what is E_0 (as opposed to E_k and $E_{k,iso}$ that are used in the text, and also not always clearly defined)? Non-standard normalisations for some of the variables (e.g., n) require the reader to do unnecessary mental arithmetic

We added a description of the afterglow parameters to the table caption.

Finally, I would have liked to see some discussion of whether there may be other events like this lurking in the archival data, or whether this was truly exceptional because of the follow-up campaign. Even if this event is exceptional, is it possible to make statistical inferences about other apparently underluminous short GRBs? Are there specific strategies to follow these up even in the absence of gravitational-wave signals?

As we mention in the introduction, we believe that it is possible that similar events escaped identification due to the lack of an optical or X-ray counterpart at early times. The total number of these events is ~30 in 13 years of Swift mission, although we believe that the number of local events is probably a small fraction of it (~10-15%). We are actually working to address the questions posed by the referee and are planning to report the results in a separate publication.

Reviewer #3 (Remarks to the Author):

Dear Authors, I have read with great pleasure your manuscript and I think it is well written, clear, and convincing in its conclusions. Definitely to be published.

I do not have really major comments, but there are moderately important issues that I think should anyway be addressed to further improve the paper.

Since the submission of the paper we have now an improved view of the late evolution of the GRB170817A (e.g. Lyman et al. 2018, arXiv:1801.02669). Although already now the discussion is fair, I would suggest to update the paper according to these new findings when you refer to GW170817.

We added a reference to that latest work of Lyman et al.

In the literature, hints of kilonova detections were already reported. I refer here at least to the papers by Tanvir et al. (2013, Nature 500, 547), Yang et al. (2015, NatComm 7, 7323), Jin et al. (2015, ApJ 811, 22) and Jin et al. (2016, NatComm 7, 12898). You mention and discuss already the last one. Now, I am well aware of the limitations on the number of references, and which papers should be quoted is entirely a choice of the authors. However, I warmly suggest to better stress that, in spite of the fact that your analysis is of great interest and meaning, there have been remarkable previous efforts to identify elusive kilonova signatures in short GRB light-curves.

We added the results from Tanvir et al. (2013) into our discussion.

One thing that appears immediately clear to the reader is the strongly different result obtained in this paper compared to the Fong et al. (2016) paper. There is nothing bad in it, however you should discuss and summarise, briefly, the main reasons for that.

We added a paragraph noting that Fong et al. (2016) only took into account the *Chandra* data and therefore proposed a standard power-law decay to describe the afterglow temporal evolution.

We believe that the lack of *Swift* data in their analysis did not allow them to catch the atypical afterglow behavior and the early optical excess.

We also used a different approach in our comparison between this GRB and the larger sample of short GRBs. Fong et al. (2016) consider the optical and X-ray luminosities of GRB150101B and find that they lie within the range of short GRBs. We agree with this statement. However, driven by the properties of GW170817, we also included the gamma-ray energy release in our comparison as a low gamma-ray energy/bright optical emission was one of the first distinctive and surprising features of its electromagnetic counterpart.

Then a few minor comments, often just typos:

. End of the Introduction. It could be interesting to also quote the percentage of Swift short GRBs with no accurate localisation.

We added this number to the text.

. 84. I do not know if you are using latex, yet it could be better to write “ $\{\rm iso\}$ ”.

We are using Word but still tried to change the format.

. 106. A factor 2 brighter?, 2 times brighter?

Fixed

. Temporal analysis section. Here, if required (only if required), you could save some space with the discussion of the AGN analysis that could be shorter.

We shortened the text on the properties of the AGN.

. 226. Could you quantify how much brighter?

We added to the text that is ~ 2 times brighter.

. 362. How does the derived density compare with what typically obtained analyzing short and long GRB afterglows?

The derived density is quite a typical value for short GRBs, although not uncommon for long GRBs either. The inferred value is indeed in the typical range of densities measured for massive elliptical galaxies.

. Gamma-ray data reduction Methods section. I think that the concept of “statically favored” fit is often ambiguous. There are, given a confidence level, acceptable or not acceptable fits. Having the smallest residuals is not enough to claim, in general, that a fit is better than another. Your following discussion is indeed correct, and therefore I would just suggest to slightly re-formulate the paragraph.

We agree and reworded the paragraph. We further expanded this section with a refined analysis of the GBM data, which led to a revised estimate of the burst duration and total energy release.

. 464. 2-pixel radius?

Fixed.

REVIEWERS' COMMENTS:

Reviewer #1 (Remarks to the Author):

The authors have fully addressed the points I raised. I recommend the publication of this paper in Nature Communications.

Reviewer #2 (Remarks to the Author):

This is the second report on "A luminous blue kilonova and an off-axis jet from a compact binary 1 merger at $z=0.1341$ " by Troja et al. (manuscript number NCOMMS-18-19107).

I would like to thank the authors for their clear and detailed responses to my queries.

I have no hesitation in recommending the revised manuscript for publication.

I attach a few follow-up comments for the authors to consider:

Around line 286: "suggestive of a high natal kick velocity for its progenitor, which merged far from its birth site" — I don't find this inference convincing (though I suppose it depends on what precisely is meant by "high"). Given that the stellar population is old (~ 2 Gyr), is it possible to rule out formation at the present site several Gyr ago? And even if the formation happened, say, 10 kpc away, a kick of only ~ 5 km/s is nominally required to traverse this distance in ~ 2 Gyr (ignoring the galaxy potential).

Around line 399: "occurs before any significant spreading takes place" — if there is sideways expansion of the jet, it would presumably occur after the jet break, i.e., when the Lorentz factor of the core of the structured jet reaches $1/\theta_c$ — well before it reaches $1/\theta_v$ and becomes visible to the observer. That would change the peak time scaling from $(\theta_v)^{8/3}$ to $(\theta_v)^2$. In addition to observer angle and energy differences, the different peak times for GW170817 and GRB150101B afterglows could also be due to different ISM densities.

Around line 424: "in the nearby ($z < 0.2$) Universe, where off-axis explosions are detectable by current gamma-ray facilities" — I thought GRB170817A wouldn't have been detectable at anywhere close to 1 Gpc ($z \sim 0.2$).

Table 2: the column labeled "Afterglow Flux" now reports optical data identified with a kilonova rather than the GRB afterglow; perhaps relabel the column to "Observed flux" or similar to avoid confusion?

Reviewer #3 (Remarks to the Author):

Dear Authors,

I am satisfied with your replies to my comments, and I am now gladly supporting the publication of your paper in Nature Communication.

Reviewer #2 (Remarks to the Author):

This is the second report on "A luminous blue kilonova and an off-axis jet from a compact binary 1 merger at $z=0.1341$ " by Troja et al. (manuscript number NCOMMS-18-19107).

I would like to thank the authors for their clear and detailed responses to my queries.

I have no hesitation in recommending the revised manuscript for publication.

I attach a few follow-up comments for the authors to consider:

Around line 286: "suggestive of a high natal kick velocity for its progenitor, which merged far from its birth site" — I don't find this inference convincing (though I suppose it depends on what precisely is meant by "high"). Given that the stellar population is old (~ 2 Gyr), is it possible to rule out formation at the present site several Gyr ago? And even if the formation happened, say, 10 kpc away, a kick of only ~ 5 km/s is nominally required to traverse this distance in ~ 2 Gyr (ignoring the galaxy potential).

We removed the adjective 'high' from the sentence.

Values inferred for galactic binary pulsars span a wide range (from a few km/s to a few hundreds km/s) and we agree that in our case there is no evidence for velocities in the upper range ($\sim > 100$ km/s). However, the observed GRB location suggests a minimum value of ~ 5 km/s, as derived from the referee. If one takes into account the typical velocity dispersion for a massive elliptical galaxy (~ 260 km/s; Bloom et al. 2006), then a more realistic estimate is $\sim 20-30$ km/s.

If, as the referee suggests, there was a significant episode of star formation at the explosion site, we would expect the GRB position to correlate with the galaxy light tracing the stellar mass. The lack of such correlation suggests that the progenitor moved away from its birth site.

Around line 399: "occurs before any significant spreading takes place" — if there is sideways expansion of the jet, it would presumably occur after the jet break, i.e., when the Lorentz factor of the core of the structured jet reaches $1/\theta_c$ — well before it reaches $1/\theta_v$ and becomes visible to the observer. That would change the peak time scaling from $(\theta_v)^{8/3}$ to $(\theta_v)^2$.

The jet does not experience significant sideways expansion until it becomes non-relativistic (e.g. van Eerten, MacFadyen, & Zhang 2010; Duffell & Laskar 2017) around Lorentz factor ~ 1.4 ($u \sim 1$). Our t_{peak} scalings are valid so long as the viewing angle is less than $< \sim 0.7$ rad, so the peak time (Lorentz factor $\sim 1 / \theta_{\text{view}}$) occurs when the blast wave is still relativistic. This system falls well within this bound. For larger viewing angles sideways expansion will certainly change the t_{pk} scaling. We added to the text that our discussion is valid for $\theta_{\text{view}} < 0.7$ rad.

In addition to observer angle and energy differences, the different peak times for GW170817 and GRB150101B afterglows could also be due to different ISM densities.

We amended the text by adding ‘for similar explosion properties’

The peak time t_{pk} is proportional to $n^{-1/3}$, where n is the density, whereas the dependence on the viewing angle is stronger ($\sim \theta_v^{8/3}$). It follows that one needs orders of magnitude variations in n in order to produce a detectable effect on t_{pk} , whereas small changes in the jet orientation affect it more prominently.

Around line 424: “in the nearby ($z < 0.2$) Universe, where off-axis explosions are detectable by current gamma-ray facilities” — I thought GRB170817A wouldn’t have been detectable at anywhere close to 1 Gpc ($z \sim 0.2$).

That is correct. GRB170817A was detected only because it happened in the local (< 100 Mpc) Universe. However, GRB150101B shows that off-axis events can be detected at cosmological distances ($z = 0.1341$ in this case). The detectability of an event depends on the intrinsic energy of the explosion, the jet geometry and the viewing angle. It has been shown (ref. 54) that this population of off-axis events can substantially contribute to the observed sample of gamma-ray bursts for $z < \sim 0.2$.

Table 2: the column labeled “Afterglow Flux” now reports optical data identified with a kilonova rather than the GRB afterglow; perhaps relabel the column to “Observed flux” or similar to avoid confusion?

We agree and replaced ‘Afterglow’ with the more generic ‘Counterpart’.